# Metal-Based Nanoparticles and Their Relevant Consequences on Cytotoxicity Cascade and Induced Oxidative Stress

**DOI:** 10.3390/antiox12030703

**Published:** 2023-03-12

**Authors:** Yunhui Min, Godagama Gamaarachchige Dinesh Suminda, Yunji Heo, Mangeun Kim, Mrinmoy Ghosh, Young-Ok Son

**Affiliations:** 1Interdisciplinary Graduate Program in Advanced Convergence Technology and Science, Jeju National University, Jeju-si 63243, Republic of Korea; 2Department of Animal Biotechnology, Faculty of Biotechnology, College of Applied Life Sciences, Jeju National University, Jeju-si 63243, Republic of Korea; 3Department of Biotechnology, School of Bio, Chemical and Processing Engineering (SBCE), Kalasalingam Academy of Research and Educational, Krishnankoil 626126, India; 4Bio-Health Materials Core-Facility Center, Jeju National University, Jeju-si 63243, Republic of Korea; 5Practical Translational Research Center, Jeju National University, Jeju-si 63243, Republic of Korea

**Keywords:** metal-based nanoparticles, reactive oxygen species, oxidative stress, inflammation, cytotoxicity, bio-medical

## Abstract

Emerging nanoscience allows us to take advantage of the improved evolutionary components and apply today’s advanced characterization and fabrication techniques to solve environmental and biological problems. Despite the promise that nanotechnology will improve our lives, the potential risks of technology remain largely uncertain. The lack of information on bio-impacts and the absence of consistent standards are the limitations of using metal-based nanoparticles (mNPs) for existing applications. To analyze the role played by the mNPs physicochemical characteristics and tactics to protect live beings, the field of nanotoxicology nowadays is focused on collecting and analyzing data from in vitro and in vivo investigations. The degree of reactive oxygen species (ROS) and oxidative stress caused by material nanoparticles (NPs) depends on many factors, such as size, shape, chemical composition, etc. These characteristics enable NPs to enter cells and interact with biological macromolecules and cell organelles, resulting in oxidative damage, an inflammatory response, the development of mitochondrial dysfunction, damage to genetic material, or cytotoxic effects. This report explored the mechanisms and cellular signaling cascades of mNPs-induced oxidative stress and the relevant health consequences.

## 1. Introduction

Throughout the twenty-first century, revolutionary nanotechnology has been rapidly expanding and intriguing scientists, medical professionals, and consumers [1,2]. Material nanoparticles (NPs) have distinct physicochemical qualities due to their nano-size (<100 nm), wide surface area, and rapid reactivity. Nevertheless, more than ~2800 commercial nanoparticulate-based applications are promptly available [3]. These NPs exhibit specific physicochemical characteristics and are manufactured for applications in biological and commercial functions, including cancer research [4], drug delivery, cosmetics, biosensors, environmental remediation, antimicrobial agents, and environmental remediation [5]. The population susceptible to metal-based nanoparticles (mNPs) is growing as their application develops. According to the allied market report (https://www.alliedmarketresearch.com/nanotechnology-market, published in July 2021), the worldwide nanotechnology market was valued at USD 1055.1 million in 2018 and is estimated to reach USD 2231.4 million by 2025, with a total value of USD 43.1 billion by 2030 with a CAGR of 18.05% from 2022 to 2030. However, with the fast growth of nanotechnology and the widespread usage of nano products, the hazard of human exposure is rapidly increasing [6]. mNPs are frequently consumed through food, drink, air, cosmetics, or pharmaceuticals and absorbed by the body before being redistributed to major organs, potentially causing tissue damage.

Despite the undeniable benefits of nanoscale materials’ potency, there are questions which still need to be solved regarding how NPs are implemented in everyday life and may or may not impact the environment. The direct and indirect effects of mNPs on the environment and air pollution may be researched from various perspectives. It may inevitably enter the environment by producing, using, and discarding nanoparticle products. These mNPs may accumulate at a certain “level to environmental risk assessment” that necessitates an understanding of the relationship due to their propensity to aggregate, agglomerate, and maybe adsorb or interact with biological matter. As a result, living creatures are constantly exposed to inorganic NPs, which enter the human body through various paths such as ingestion, inhalation, dermal penetration, and blood circulation [7] and are translocated in organs (liver, kidneys, spleen, heart, and brain) and tissues, depending on their physicochemical properties (Figure 1). To assess the threat of inorganic NPs, it will be crucial to comprehend the precise mechanism underlying their cell-specific cytotoxicity and the validity of toxicity analytical techniques.

The current focus of nanotoxicology is on gathering and interpreting data from in vivo and in vitro studies to ascertain how the physicochemical properties of NPs and mitigation strategies influence living beings. These variables affected the biokinetics and biological activity of NPs, influencing their translocation from epithelia into organs, intracellular localization, the stimulation of ROS generation, and interaction with receptors [8]. In this study, we discussed the mechanisms and cellular signaling cascades of mNPs-induced oxidative stress and their possible detrimental impacts on health. A comprehensive search was undertaken using relevant electronic databases such as PubMed, Scopus, Google Scholar, Web of Science, ProQuest, ERIC, WorldCat Discovery, JSTOR, and Directory of Open Access Journals (DOAJ). The search followed a strategy to select literature that addressed the review objectives.

## 2. Revolutionary Implications of mNPs in Pharmaceutical Research

The essential features of mNPs are their size, which lies in the transitional zone between individual atoms or molecules and comparable bulk materials [9,10]. The engineered physicochemical properties of mNPs allows them to create the opportunity for an increased uptake and interaction with biological tissues. Because of these features, mNPs have potential uses in biological diagnostics, cell labeling, targeted drug administration, medical imaging, cancer treatment, and biological sensors. They have also been employed as a gene carrier for selective tissue or cell distribution [11], transfection vectors [12], and contrast agents in medical imaging fluorescent labels [13]. mNPs comprised of titanium dioxide (TiO2), copper (Cu), iron (Fe), zinc (Zn), aluminum (Al), gold (Au), and silver (Ag) are also gaining popularity.

It is believed that the quantity of silver that individuals consume everyday by consumption is about 20–80 µg [7]. Numerous studies have been conducted on broadly used silver NPs (AgNPs) in healthcare as potential antimicrobial agents against more than ~650 different types of disease-causing organisms, including viruses, biocides, antibiotic treatment alternatives, and nanocomposite coatings [2,14,15,16,17]. Additionally, interactions between AgNPs and sulfur-containing proteins can impact the ability of bacteria to survive. AgNPs can impair cellular respiration and cell death by exchanging other essential metal ions from bacterial cells, such as Zn^2+^ and Ca^2+^ [18]. Furthermore, gold NPs (AuNPs) are employed to assault bacterial membranes and disrupt DNA replication, killing certain bacteria, including *E. coli, Staphylococcus aureus*, and *Pseudomonas aeruginosa* [19,20,21,22]. Peroxidase-like nanoparticles, e.g., AuNPs, can be employed as labels in various analytical techniques, such as virus detection. The immunochromatographic strip nanozyme-strip made of Fe_3_O_4_ magnetic NPs can identify the Ebola virus’ glycoprotein. The authors suggest that it may be feasible to detect additional pathogenic viruses, such as the Bunya virus [23]. Concerns regarding conflicting bio-effects are raised by the dual function of ZnO and TiO_2_ in protecting skin from sunlight and eradicating surface microorganisms, respectively [2].

The use of NPs in drug delivery is a potential technique for improving the administration of chemotherapies, radionuclides, and antibody medicines to target cells. Al NPs have been proposed as drug delivery systems, mainly by encapsulating medications with aluminum–magnesium hybrids to boost solubility, prevent clearance processes, and allow for the site-specific targeting of medications to cells. Anticancer medications delivered with zinc oxide and gold NPs are thought to have an antitumor effect by stimulating the immune system [24]. Research demonstrated that adding ZnO NPs to doxorubicin boosted the drug’s intracellular concentration in hepatocarcinoma cells, increasing its therapeutic impact compared to controls [25]. Recent advances in nanotechnology have made it feasible to combine NPs with biological components for a targeted treatment, enabling the administration of radiation doses with a greater specificity and limiting harm to healthy tissues [22].

The commercial applications of nanotechnology in medical research are mostly focused on the production of nano-agents for labeling, tissue engineering, and medication delivery systems [26]. Studies have primarily focused on cellular imaging with the imaging of in vivo macrophage activity. The fabrication of mNPs employed as markers in medical diagnostics, such as iron oxide (Fe_2_O_3_) or superparamagnetic-based nanocrystal technology and other noble metal plasmon-resonant particles (i.e., Au, Ag, Pt, and Pd), have dramatically increased in medical imaging approaches, such as magnetically targeted drug delivery [27], MRI contrast agents [28], magnetically assisted gene transfection [29], triggered drug release [30], and magnetic fluid hyperthermia (MFH) [31,32,33]. These NPs must have high magnetization values, be smaller than 100 nm, and have a narrow particle size distribution for all these biological and bioengineering applications [32]. PdNPs have been employed in medical diagnostics, such as glucose detection, because of their enzyme-like function. Ru, Au, and Pt nanoparticle-based biosensors were utilized to detect bilirubin, ultra-low levels of mercury, catalase (CAT) and superoxide dismutase (SOD), polyphenol oxidase, ferroxidase, and ascorbate oxidase mimics, accordingly [34].

The nanowires and nanospheres of iron, gadolinium, and NiO have shown excellent imaging signals [35]. It has been reported that 198Au NPs may be synthesized directly involving radionuclides [36]. Xie et al. described the process for radiolabeling gold nanoshells with the radionuclide labels copper-64 (64Cu) and indium-111 (111In) through a bifunctional PEG and chelating agent in live rats. They demonstrated head and neck squamous cell carcinoma xenografts using non-invasive PET and SPECT imaging [22,37]. These engineered mNPs-altered physicochemical and structural characteristics may cause various material interactions and toxicological consequences [38,39]. Transition metals, including Fe, Ni, Cu, and Cr, may be absorbed into the surface of mNPs due to their enormous specific surface area and catalyze Fenton reactions that directly damage DNA [40], which may lead to carcinoma. Hence, to prevent certain metals from leaching, non-permeable coatings are required.

## 3. Physio-Pathological Implications of mNPs

mNPs are extremely attractive in various applications due to their distinctive physicochemical characteristics, allowing them to influence cellular processes at the biological level [41,42]. The fact that mNPs have high surface-to-volume ratios makes them reactive or catalytic [1]. Due to their small size, they are more likely to be able to penetrate biological barriers such as cell membranes and cause cellular dysfunction in living organisms [42]. Indeed, the high toxicity of some transition metals can make it challenging to use mixed oxide NPs in biomedical uses. It triggers adverse effects on organisms, causing oxidative stress, stimulating the formation of ROS, mitochondrial perturbation, and the modulation of cellular functions, with fatal results in some cases [39,43,44] (Figure 2).

Size-dependent cellular interactions with known physiologically active silver nanoparticles reported a strong inflammatory response by emitting tumor necrosis factor-α (*TNF-α*), MIP-2, and interleukin-1β (*IL-1β*), and the oxidative stress might be the principal cause of cellular damages [45].The primary sources of mNPs-induced oxidative stress are (i) mNPs or transition metal pollutants employed as catalysts in the manufacture of nonmetal nanoparticles; (ii) intermediates of highly stable free radicals found on “reactive” surfaces of particles such as quartz and carbonaceous particles; and (iii) the functionalization of NPs to produce redox-active groups [6,46,47,48]. Since oxidative stress is a significant factor in NP-induced injury, it can result in a variety of physiopathologic effects, including genotoxicity, necrosis, inflammation, fibrosis, metaplasia, hypertrophy, lipid peroxidation, damage to macromolecules such as DNA, leading to mutations, and fostering the growth of tumors [49,50,51]. It is crucial to comprehend the underlying processes of nanoparticle-induced ROS and the accompanying oxidative stress they impose.

### The Adverse Impacts of Various mNPs on the Organs

Since mNPs are used increasingly often in healthcare, the safety of employing them for the central nervous system (CNS) is receiving increased attention [52] (Figure 3). This CNS of vertebrates is separated from the rest of the body by the blood–brain barrier (BBB), responsible for the exchange of nutrients and metabolites between the blood and brain, xenobiotics penetration prevention, and restricting immune cell infiltration [53]. Several studies demonstrated the transport of various mNPs, e.g., ultrafine Ag NPs [54], Al NPs [55], CdSe/ZnS, quantum dots (CdSe/ZnS QDs) [56], copper oxide NPs (CuO NPs) [57], ultrafine manganese dioxide NPs (MnO_2_ NPs) [58], titanium dioxide NPs (TiO_2_ NPs) [59], and aluminum oxide NPs (Al_2_O_3_ NPs) [60,61], leading NPs (Pb NPs) [62] to the brain via the olfactory nerve, which is a direct route, circumventing the BBB [63]. After being exposed to Ag NPs, microvascular endothelial cells, astrocytes, and pericytes showed mitochondrial shrinkage, endoplasmic reticulum expansion, and vacuolation. There have also been alterations in 23 genes related to cell death, metabolic and biosynthetic processes, and response to stimuli [64]. Because of their outstanding biocompatibility and simple biodegradation in vivo, magnetite-based iron oxide NPs (below 30 nm) have gathered the most interest for various diagnostic and potential therapeutic applications in the CNS. In addition, after being metabolized, iron ions from these NPs are incorporated to the iron deposits or become absorbed by erythrocytes as part of the hemoglobin. It has been described that ultrasmall superparamagnetic iron oxide particles can cross the human BBB [65]. Hence, it is essential to precisely characterize the neurotoxic effects of mNPs in order to promote the creation of safer and more effective CNS-targeting nanomedicines.

Another concern is the influence of mNPs on offspring. Until now, studies have shown that NPs entering the maternal body during gestation may damage fetal development through direct or indirect processes. According to neurodevelopmental research, prenatal damages by mNPs lead in distinct phenotypes in both male and female offspring and influenced the expression of functional genes related to brain development in mice [66,67]. According to a study, subcutaneously injected TiO_2_ NPs into pregnant mice caused translocation to the offspring and had an adverse effect on the male offspring’s genital system by lowering the daily sperm production [66,68]. Additionally, it was shown that adult mice that inhaled TiO_2_ NPs experienced lengthy lung inflammation and, as a result, their progeny displayed aberrant neurobehavioral abnormalities [69].

Researchers have recently emphasized the potentially hazardous cardiovascular repercussions of exposure to mNPs, particularly engineered nanoparticles, due to their extensive applications in electronics, mechanical design, environmental remediation, and biomedicine (Figure 3). According to a study by Kan et al., there are three potential mechanisms for adverse effects on the cardiovascular system: (a) synthetic NPs may travel from the lungs to the bloodstream and cause pathological alterations in the tissue of the heart and arteries; (b) engineered NPs may cause a systemic oxidative stress/inflammatory reaction through the lungs that changes cardiovascular function; or (c) they could affect cardiovascular performance through the neurogenic pathway [70]. Au-NPs are recognized to have antioxidant potential in therapeutic strategies; nonetheless, in vitro studies have shown that Au-NPs were associated with autophagy when they entered rat heart muscle cells. The effects of Au-NPs on the heart are proportional to their size. The study found that Au-NPs with a diameter of 40 nm were harmful to the heart. Eventually, 5 nm Au-NPs exhibited no cardiac injury. As a result, larger Au-NPs (greater than 40 nm) are more likely to induce cardiac dysfunction [71]. The Impacts of Fe_2_O_3_ NPs (spherical; 50 nm particle size; 50–245 m^2^/g surface area) and AgNPs (spherical; 50 nm particle size; 5.0 m^2^/g surface area) on the heart and lungs of male rats showed toxicity when administered either alone or in combination [72]. Cardiotoxicity and lung toxicity were triggered through JNK, p53, and NF-κB pathways, which included oxidative DNA modification, the activation of inflammation, free radical production, and the inhibition of antioxidant defense. mNPs caused distinct metabolic changes that may have important implications as atherogenic causes. However, PON1 is a hydrolytic enzyme with a broad substrate range that can protect against lipid oxidation. PON1 was depleted in the plasma and cardiac tissues by mNPs, such as Fe_2_O_3_NPs and AgNPs [72]. The toxicity of TiO_2_ and AgNPs on Daphnia magna revealed that the heart rate dropped with increasing concentrations and that the ROS levels were elevated in comparison to the control group [73]. In order to effectively implement mNPs in a therapeutic context, it is necessary to efficiently measure their preventive efficacy in relevant cardiac pathological conditions.

Several mNPs have been demonstrated to cause hepatic steatosis due to residual NPs in the liver [74]. Despite the fact that 30–99% of specified NPs will accumulate and sequester in the liver, the accumulation of specific mNPs in the liver has been observed to induce oxidative stressors, which in turn disrupt the liver’s metabolism and homeostasis [75,76]. In vivo pulmonary toxicity experiments were performed using five distinct particle types, including (1) carbonyl iron (CI), (2) crystalline silica (CS) (Min-U-Sil 5, α-quartz), (3) precipitated amorphous silica (AS), (4) nano-sized zinc oxide (NZO), or (5) fine-sized zinc oxide (FZO). This study revealed that instilled carbonyl iron particles caused slight toxicity, whereas crystalline quartz silica particle exposure caused sustained inflammation and cytotoxicity, and amorphous silica particle exposure caused transient inflammatory responses that were reversible one week later. Intriguingly, the intratracheal intravenous administration of nano or fine-sized zinc oxide particles elicited strong but transitory inflammation/cytotoxic effects that were reversed by exposure one month after instillation [77,78]. Jia et al. demonstrated that the oral administration of 10 or 20 nm Ag NPs in normal mice promoted the progression of fatty liver disease from steatosis to steatohepatitis in obese mice, which can be attributed to the pro-inflammatory activation of KCs in the liver, increased hepatic inflammation, and the suppression of fatty acid oxidation [79]. It was reported that 14 ± 4 nm Au NPs with spherical forms were documented to induce hepatic steatosis in Wistar-Kyoto rats [80]. In contrast, 60 nm spherical Si NPs exacerbated hepatic steatosis in mice or a zebrafish model via the TLR5 signaling pathway, while SiO_2_ NPs caused oxidative damage and triggered the transforming growth factor-β1 (TGF-β1)/Smad3 signaling pathway, which accelerated the liver fibrosis process [81,82]. In addition to aggravating the liver, Fe_2_O_3_ NPs cause higher ALT levels, increased hepatocyte necrosis, hepatic inflammation, interstitial congestion, and fatty degeneration near the central vein [74], whereas Fe_3_O_4_ NPs cause higher ALT (alanine transaminase), AST (aspartate aminotransferase), and ALP (alkaline phosphatase) levels, central venous congestion, hepatocyte hypertrophy, regeneration, and necrosis, as well as increased Kupffer cell counts [83]. Furthermore, the co-administration of ZnO NPs with the xenobiotic chemical organophosphate dimethoate-enhanced hepatic deposition of zinc and dimethoate, led to increased liver oxidative stress and damage [84].

The kidneys are essential organs that filter all hazardous chemicals and metabolites through urine. Nonetheless, the key diseases that contribute to kidney disorders are oxidative stress, inflammation, apoptosis, and necrosis. However, the effects of NPs to address kidney dysfunction has not been extensively studied in in vivo and in vitro animal models. Most researches have pointed to the significance of ROS in SiNPs toxicity; as a result, SiNPs evoked an inflammatory response in macrophages and the kidney of mice [85,86]. The administration of mesoporous silica NPs (mSiNPs) increased the expression of NF-κB, MyD88, TLR4, caspase-3, and p65 in the liver and kidney of rats, along with the levels of ROS, lipid peroxidation, and nitric oxide, while suppressing antioxidants and Nrf2/HO-1 signaling [87]. The toxicity profile of nano-sized TiO_2_ delivered to mice has also been described in the lung, liver, and kidney [7,88,89]. Some important studies that have shown the toxic effects of AgNPs on different cell lines, including embryonic kidney cells (HEK293T) and porcine kidney cells (Pk 15) [90,91]. According to a previous study, AgNPs with diameters less than 100 nm are mostly taken up by endocytosis in epithelial cells and can cause oxidative stress, DNA damage, and inflammation in enterocytes [92]. Kidney damage inflicted by various AgNP doses was also studied on male Wistar rats after 28 days of oral administration [93]. A histological examination indicated damage to the lining tubular epithelial cells after treatment with 30 and 125 mg/kg of AgNPs, including vacuolization, hazy swelling, severe necrosis, and pyknotic nuclei, confirming the toxicity induced by these doses [93].

## 4. Effects of mNPs on Cytotoxicity and Cellular Damage

The most pressing worry is how hazardous mNPs are at regular doses [94]. Given the unique nature of nanoparticles, it is difficult to link the test results from diverse studies and evaluate whether the toxicity and cellular damage seen are physiologically significant. The cytotoxicity of mNPs is being studied using a variety of standardized techniques, including in vitro and in vivo studies, involving rats, humans, and aquatic species such as zebrafish, catfish, algae, and macrophages, as well as comprehensive genomic or biodistribution studies. The inherent toxicity of specific mNPs may alter biological behavior in terms of proteins, cells, subcellular structures, tissue, and organs [45,95,96,97,98,99,100,101]. Exposure to these mNPs is associated with oxidant production, macrophage activation, the extended release of inflammatory mediators and growth factors, and the fibroblast stimulation of the creation of the extracellular matrix at the cellular level.

An in vivo study on gastrointestinal persorption and the tissue distribution of differently sized colloidal gold NPs revealed that it could permeate the small intestine and spread to the blood, brain, lung, heart, kidney, spleen, liver, intestine, and stomach [102]. It was found that mNPs (such as AuNPs, AgNPs, and CuNPs) may not be detected by normal phagocytic defenses and cluster first outside the cell. Then, they entered the cell in an aggregated condition [94]. The cell was exposed to spherical gold NPs of varying sizes for 24 and 48 h, indicating that they are not fundamentally harmful to human cells [103]. The study by Goodman et al. stated that AuNPs are non-cytotoxic and reduce the levels of potentially hazardous ROS in the cells [104]. However, it is necessary to distinguish between cytotoxicity and cellular damage. The theories stated that NPs that exhibit little or no cytotoxicity might be capable of causing significant cellular damage indeed [94,105].

The report also revealed that the toxicity of Au NPs directly related to their shapes [102,103,106], coat on the mNPs, and surface charge [104]. The study has demonstrated that the cellular response is based on particle size, with 1.4 nm particles mainly causing fast cell death by necrosis within 12 h. In contrast, closely comparable particles 1.2 nm in diameter primarily induce planned cell death through apoptosis [107]. Gold nanoparticles’ cellular absorption and cytotoxicity has been widely researched in human leukemia cells and skin HaCaT keratinocytes. CTAB-coated Au nanorods were shown to be more hazardous to human HaCaT keratinocytes than spherical Au NPs (30 nm) [94,103]. According to another study, Au NPs did not trigger cytotoxicity in human astrocytes, but they instead boosted ROS generation, upregulated NF-kB activity, and decreased micronuclei development [108].

Ag NPs have been widely used as antimicrobials; however, the use of ~25 nm Ag NPs (at concentrations of >25 µg/mL) in the image analysis of neural tissue and cells, in particular, raises concerns about the possibility that they may be contributing to neurodegenerative diseases (such as Parkinson’s and Alzheimer’s), due to their capacity to produce ROS and oxidative stress [2,109]. The dose-dependent Ag NPs-induced toxicity experiment was carried out in zebrafish embryos [110]. An in vivo dose-dependent investigation of Ag NPs using adult Sprague Dawley rats revealed substantial alterations in plasma alkaline phosphatase (ALP) and blood cholesterol, indicating that these NPs might harm the liver [111]. In PC12 cells, Ag NPs enhanced ROS generation and upregulated the expression of genes associated with oxidative stress, including those encoding HO-1 and MMP-3 [112]. An incident involving a 71-year-old man who consumed colloidal silver every day for four months was reported to have developed myoclonic status epilepticus and fallen into a coma. It was observed that the patient’s plasma, erythrocytes, and CSF all had high quantities of silver [113].

Copper NPs (CuNPs) have also received significant attention as potential candidates for innovative antimicrobial applications such as biocides, antibiotic treatment alternatives, and nanocomposite coatings [114]. However, they have severe toxicological consequences through oxidative damage and/or an inflammatory reaction [115]. Aside from industrial-scale uses, transition metal oxide particles have revolutionized several sectors, including catalysis, sensors, optoelectronic materials, and drug delivery. The study suggested that transition metals, including Si, Zn, Cu, Fe, chromium (Cr), and vanadium (V), are linked to ROS production via Fenton and Haber–Weiss reaction pathways [46]. The toxicity of metal oxide nanoparticles, such as nano-TiO_2_ [116], nano-ZnO [117], nano-CuO [118], nano-Fe_3_O_4_ [119], Al_2_O_3_, and CrO_3_ of particle sizes ranging from 30 to 45 nm, is widely reported [120,121]. TiO_2_ toxicity studies have indicated that they cause inflammatory responses, cell death, an increase in ROS, and the stimulation of oxidative stress-related genes [122,123,124]. It has been demonstrated that these NPs interact directly with the liver and, as they migrate through the organ systems, might cause synaptic damage and neurotransmitter dysfunction called neurotoxicity [2,125,126]. According to the reports, if these NPs can efficiently distribute to specific tissues by penetrating the epidermis, entering via damaged skin, or being administered systemically, they may threaten the overall health of the body [127,128,129]. According to the study, nano ZnO can influence the diversity, metabolism, and functional pathways of the human gut microbiome and the gut resistome [117]. Considering the risk mentioned above, NPs may enter the bloodstream or come into direct contact with the respiratory system, which could lead to endothelial cell membrane toxicity; more even, transfer to the lymphatic system may result in secretory immunological responses.

Environmental factors such as air pollution can cause oxidative damage in the brain, which may result in neurodegenerative illnesses. NPs disrupt the tight junctions of the blood–brain barrier or enter the central nervous system (CNS), allowing them to access and induce neurotoxicity [130,131,132,133]; however, how NPs induce oxidative stress in the brain remains unclear. In vivo experiments on rats’ brains demonstrated that oxidative stress caused by TiO_2_ NPs may cause hippocampus apoptosis and spatial recognition memory impairment [134]. After being exposed to TiO_2_ NPs, the concentration of Ti in the mouse brain elevated, causing an increase in ROS generation, inhibition of antioxidant activities in hippocampal regions, and an increase in the proportion of apoptotic cells [135]. It appears to be a paradox that different types of mNPs induce a broad range of adverse effects in humans. Despite the fact that at the cellular/molecular level, mNPs have been found to alter analogous pathways and processes, the majority of which are based on oxidative stress.

### 4.1. Influence of mNPs on Excessive ROS Generation

ROS serve as cell signaling agents for typical biological activities; nevertheless, the overproduction of ROS can induce oxidative stress, resulting in damage to several cellular organelles and functions, which can eventually cause disrupt normal physiology [136,137,138] as well as significant genotoxicity (Figure 4). The cellular oxidative stress is initiated by an imbalance between the generation of ROS (including the superoxide radical, hydroxyl radical, hydrogen peroxide, nitric oxide, and peroxynitrite) and lessened antioxidant defenses inside the cell, which can cause a wide range of diseases [139]. The toxicity of mNPs was assessed utilizing mitochondrial and cell membrane viability, as well as ROS, which revealed a dose-dependent reduction in cell viability. ROS can cause membrane damage, lipid denaturation, and DNA structural changes. This increases ROS levels in the mitochondria, lowering ATP that causes a flux in the tricarboxylic acid (TCA) cycle and the reduction in cardiolipin. There is evidence to suggest that mitochondria-derived ROS may be involved in the apoptosis that is brought on by *TNF-α* and *IL-1β* [140,141,142,143]. Excessive ROS generation triggers a cascade of proinflammatory cytokines and mediators through redox-sensitive MAPK and NF-B signaling pathways, which control the transcription of inflammatory genes including *IL-1*, *IL-8*, and *TNF-α*. Moreover, ROS directly regulate neuronal ion channels, kinases, and transcription factors, which is an essential aspect in brain development [144,145,146]. A lot of studies have found that ROS play an important role in neurodegenerative illnesses including Alzheimer’s and Parkinson’s disease.

The physical and chemical characteristics of mNPs, such as their size, chemical composition, surface area, and charge, can impact the mNPs-mediated ROS formation mechanism. The primary driving mechanisms underlying NP-induced ROS include prooxidant functional groups on the reactive surface, active redox cycling induced by transition of mNPs, and particle-cell interactions [46,147,148]. The nanoscale dimensions can enhance and change the electrical characteristics of the NP surface, producing reactive groups. The observation stated that cells exposed to 50 g/mL of 15 nm Ag NPs experienced a more than 10-fold rise in ROS levels implying that the toxicity is most likely caused by oxidative stress [2]. Typically, oxidative stress inhibits antioxidant enzymes such as catalase, superoxide dismutase, and glutathione peroxidase while depleting non-enzymatic antioxidants such as vitamin C, vitamin E, and glutathione [149,150].

### 4.2. Impact of mNPs-Induced Oxidative Stress

The mNPs could function as oxidants and frequently cause oxidative stress in biological systems. These mNPs can drive redox processes, which results in endogenous ROS production, leading to cause serious health hazards such as even genotoxicity. Several redox signaling pathways in cells are made up of signaling molecules like kinases and transcription factors that are negatively controlled by sensor proteins. According to conventional understanding, oxidative stress can trigger such cellular signaling by changing sensor protein thiol groups [151]. Due to their high surface-to-volume ratio, NP is reactive and susceptible to environmental stressors. After exposure to mNPs, alveolar macrophages (AM) and neutrophils triggered oxidative stress in the lungs. The immune system’s phagocytic cells, such as neutrophils, activate NADPH oxidase to produce ROS [152].

The impact of metal oxide NPs via airborne pollutants on the respiratory system was studied in vitro by exposing airway epithelial (HEp-2) cells to silicon oxide (SiO_2_), ferric oxide (Fe_2_O_3_), and copper oxide (CuO) nanoparticles. In comparison, CuO exposure resulted in a significant increase in the levels of 8-isoprostanes and the ratio of GSSG to total glutathione, which showed that the ROS it produced caused oxidative stress in HEp-2 cells [153]. In brief, the prooxidant attributes of copper NPs enable them to either inhibit antioxidants or produce more ROS, which increases oxidative stress. The Cu NPs inhibit cellular antioxidant enzymes like catalase and glutathione reductase while increasing glutathione peroxidase activity. This shows that copper NPs produce ROS and inhibit cell redox balance. The Cu ion’s toxicity causes severe liver damage, increased endoplasmic reticulum stress, and inducted neuronal apoptosis [154,155,156].

Iron NPs combine with hydrogen peroxide to form hydroxyl ions and redox-active iron, which releases hydroxyl radicals through the Fenton reaction. These free radicals may damage biological macromolecules and cell organelles and have been linked to various diseases. The iron NPs also promoted cytotoxicity by increasing LDH levels [157]. Due to excessive oxygen consumption, inadequate antioxidant defense, and an abundance of oxidation-sensitive lipids, the brain is particularly susceptible to oxidative damage. Neurons, unlike other damaged tissues, cannot be restored through regeneration. Moreover, neurotoxicity and increased protein aggregation were observed in mice with neuron-specific suppression of key autophagy proteins (i.e., Atg5, Atg7, and beclin-1) [158]. The activation of various pathways, such as the p38 member of the mitogen-activated protein kinases (MAPKs) family, which is directly impacted by iron accumulations and oxidative stress and could be the underlying cause of Alzheimer’s and Parkinson’s illnesses [159].

In particular, the Fenton-type reactions generate free radicals and interact with cellular macromolecules to cause oxidative stress. Fenton reactions typically involve a transition metal ion reacting with H_2_O_2_ to yield ^•^OH and an oxidized metal ion that is extremely reactive and toxic to biological molecules [143]. On the other hand, the Haber-Weiss reaction involves a reaction between an oxidized metal ion and H_2_O_2_ to generate ^•^OH. The brief investigation has shown that these free radicals produced directly or indirectly are activating the mitogen-activated protein kinase pathways. Indeed, the activation of pathways is mediated by an increase in the number of inflammatory mediators such NF-κB, signal transducer and activator of transcription (STATs), mitochondrial malfunction, and intracellular calcium [160]. Moreover, lipid hydroperoxides are produced as a result of the oxidation of polyunsaturated fatty acids at initial progress in generating ROS. Prooxidant metals like Cu and Fe subsequently react with these lipid hydroperoxides to produce DNA-damaging end products, including malondialdehyde (MDA) and 4-hydroxynonenal, which function as inflammatory mediators and carcinogenesis risk factors [161,162].

The studies demonstrated how mNPs affect oxidative DNA damage and gene expression, which can result in the development of tumors and/or have an impact on fertility. It has been shown that exposure to metal oxide NPs results in DNA fragmentation and the formation of oxidation-induced DNA adducts [160,163]. In human embryonic lung fibroblasts, it has been demonstrated that AuNPs (20 nm in size) at 1 nM concentration cause DNA damage via the formation of 8-hydroxyl-2′-deoxyguanosine (8-OHdG) DNA adduct, which is accompanied by a decreased expression of DNA repair and the cell cycle checkpoint genes *MAD2, BRCA1, Hus1, ATLD/HNGS1, AT-V1/AT-V2, cyclin B1*, and *cyclin B2* [164]. The ionic copper (Cu^2+^) cytotoxicity has been correlated to DNA damage and apoptosis-mediated cell death. According to a study, *Mytilus galloprovincialis* short-term exposure to CuO NPs causes oxidative stress, further leading to genotoxicity and cancer development [165].

### 4.3. Effects of mNPs on Cellular Signaling and Immune Response

Excessive ROS production and oxidative stress can activate nuclear respiratory factor (NRF) 1 and 2 via AKT (Protein Kinase B), estrogen receptor (ESR) 1, and alter proteins function implicated in the stress response pathway such as SODs, CAT, GR, GPXs, PDI, and PRDXs [166]. The release of pro-inflammatory mediators through the cytokine cascades, and activation of signaling pathways including the NF-κB, MAPK, and interferon regulatory factor 3 (IRF3), and PI3-K pathways, indicating a reciprocal relationship between oxidative stress and inflammation (Figure 5). These active transcription factors stimulate the production of inflammatory mediators as TNFα, interferons (IFNs), IL1, nitric oxide (NO), and tumor growth factor (TGF) B3, which intensify inflammation. Through increased production of cytokines like ILs, kinase activation, and phosphatase inhibition, cells are known to suppress the overexpressed oxidative stress response, which affects the phosphorylation cascade.

Protein tyrosine phosphatases (PTP) have highly reactive cysteine residues that are susceptible to oxidative stress from H_2_O_2_, free radicals, or changes in the intracellular thiol/disulfide redox state. The mNPs, such as Mg^2+^, Zn^2+^, and V^5+^, may play a crucial role in the redox control of PTP by inhibiting MAPK and EGFR [167,168,169]. The effects of uncoated AuNPs and inflammatory response in rats have shown the modulation of *IL-1β*, *IL-6*, and *TNF-α* expression [170]. The subacute exposure to 20 nm ZnO-NPs affected the immune system in juvenile and adult BALB/c mice has shown an increase of *IL-6*, *IFN-γ*, *TNF-α*, and ROS in the adult mouse; however, the same molecule levels in juvenile mice did not change significantly [171].

Mitogen-activated protein kinase (MAPK) are serine-threonine protein kinases that comprise the stress-activated MAPK, c-Jun NH2-terminal kinases (JNK), and p38 MAPK. Studies on in vivo nanotoxicity using the model organism *C. elegans* revealed that excessive ROS production could activate MAPK pathways through the inhibition and/or degradation of MAPK phosphatases and produces free radicals that can lead to the oxidative modification of MAPK signaling proteins [172]. The studies also revealed that PMK-1, p38 MAPK, and hypoxia-inducible factors are highly expressed [173,174]. Additionally, it has been demonstrated that several metal oxide nanoparticles, including those made of iron, cadmium, silica, and zinc, cause the inflammatory cytokines NF-κB induces [175,176,177]. The inhibitor of κB (IκB) degrades during oxidative stress and activates NF-κB, which subsequently translocate into the nucleus to control the transcription of its target genes [178,179].

## 5. Conclusions and Further Prospective

The mNPs have a substantial beneficial effect due to their vast array of uses in the healthcare, cosmetics, and industrial sectors. Despite the prospective benefits of employing NPs in various applications, the potential health risks connected with human exposure to these mNPs have yet to be extensively reported or understood. This systematic review was presented to explore some of the critical aspects that impact the assessment of health and safety concerns associated with mNPs exposure. According to both in vivo and in vitro studies, an attempt was made to provide an overview of the cellular mechanisms of mNPs-induced oxidative stress and the adverse effects which can eventually cause disrupt normal physiology as well as significant genotoxicity. An approach called “green” synthesis of mNPs from plant extracts must be prioritized in hopes of reducing the cytotoxicity [180,181]. mNPs display significantly varied oxidative stress pathways depending on their size, surface chemistry, and structure. Thus, surface modification of nanoparticles, modified by binding different molecules to their structure to produce NPs may potentially lower nanotoxicity, however this remains under research.

## Figures and Tables

**Figure 1 antioxidants-12-00703-f001:**
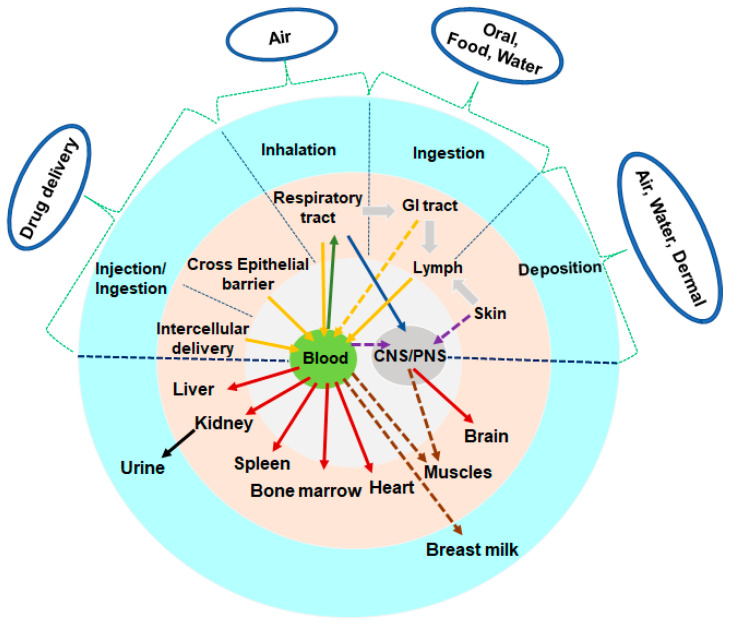
Nanoparticle and its bio interaction. The physicochemical characteristics of the surface and core of nano-sized particles will mostly determine the potential harmful consequences. Though several absorption and translocation pathways have been demonstrated, others remain hypothetical and must be studied.

**Figure 2 antioxidants-12-00703-f002:**
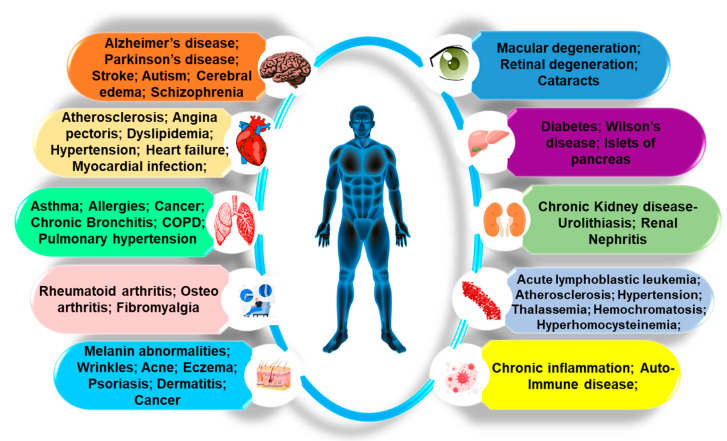
The overall negative impacts of mNPs on the human body. mNPs are oxidants that typically induce oxidative damage. mNPs can drive redox processes, resulting in endogenous ROS generation and significant health risks such as mutagenicity. (Image sources: skin anatomy, blood cells flow images by macrovector on Freepik; liver image by Freepik; arthritis image and infection image by Storyset on Freepik; human body image from brgfx on Freepik; the other images were downloaded from Pixabay). COPD—chronic obstructive pulmonary disease.

**Figure 3 antioxidants-12-00703-f003:**
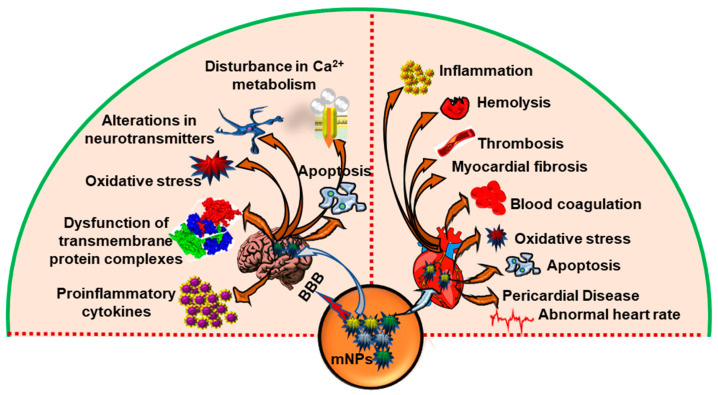
The systematic illustration of mNPs induced effects on heart and brain. Overview of the neurotoxic effects of metal based nanoplastics. mNPs can enter the systemic circulation, and eventually the brain, via penetrating the blood–brain barrier (BBB), and may cause oxidative stress and potentially cause cellular damage and neuroinflammation. The possible impact of mNPs on cardiovascular performance. Synthetic NPs may enter the circulation from the lungs and produce pathological changes in the heart and artery tissues. Moreover, this causes a systemic oxidative stress/inflammatory response via the lungs, altering cardiovascular function.

**Figure 4 antioxidants-12-00703-f004:**
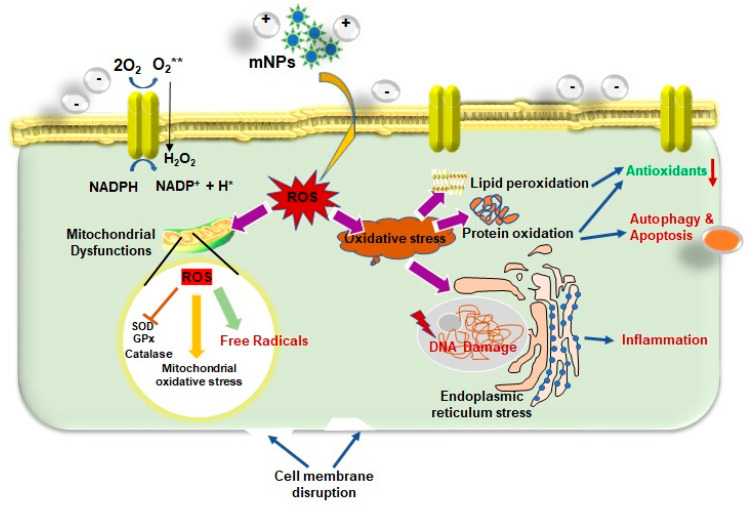
A schematic representation of the numerous triggers involved in forming reactive oxygen species (ROS) and the ROS induced pathways that lead to cell damage. Oxidative stress has a significant impact on nanotoxicity. The active surface of the nanoparticle, the size of the nanoparticle, photoactivation, toxins, metal ion dissolution, and nanoparticle interactions with biomolecules are all features that contribute to the generation of nanoparticle induced ROS.

**Figure 5 antioxidants-12-00703-f005:**
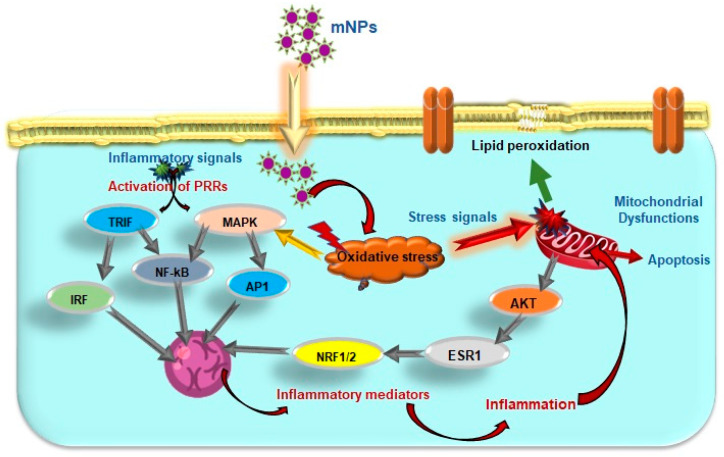
Cellular pathways connecting excess ROS production, oxidative stress and inflammation. The activated PRRs trigger downstream signalling pathways and activate transcription factors. (Image source: mitochondria and nucleus images from brgfx on Freepik).

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
