# Peer review of "Metal-Based Nanoparticles and Their Relevant Consequences on Cytotoxicity Cascade and Induced Oxidative Stress"

_antioxidants, 2023, doi:10.3390/antiox12030703_

Round 1
Reviewer 1 Report
The manuscript antioxidants-2241392 needs major revision.
The toxicity of metal-based nanoparticles depends on many factors: preparation method, size and shape of nanoparticles, surface properties, concentration, etc. The authors should make a comparison between the different types of metal-based NPs, from recent studies. In many cases, metal and metal oxide nanoparticles (especially the biosynthesized ones) show antioxidant, antimicrobial, and anticancer activities.
The authors stated that: "Furthermore, activation of the complement cascade in response to liposomes and other lipid-based nanoparticles causes hypersensitivity responses and anaphylaxis [125,126]."
But, liposomes are biocompatible and they are frequently used in biomedical fields, in cosmetics, in food industry, etc.!
I recommend updating the bibliography and introducing the following bibliographic references related to metal nanoparticles and liposomes:
1) Alemán, A.; Pérez-García,S.; Fernández de Palencia, P.; Montero, M.P.; Gómez-Guillén,M.d.C. Physicochemical, Antioxidant, and Anti-Inflammatory Properties of Rapeseed Lecithin Liposomes Loading a Chia (Salvia hispanica L.) Seed Extract. Antioxidants 2021, 10, 693. https://doi.org/10.3390/antiox10050693
2) Gorshkova, Y.; Barbinta-Patrascu, M.-E.; Bokuchava, G.; Badea, N.; Ungureanu, C.; Lazea-Stoyanova, A.; Răileanu, M.; Bacalum, M.; Turchenko, V.; Zhigunov, A.; Juszyńska-Gałązka, E. Biological Performances of Plasmonic Biohybrids Based on Phyto-Silver/Silver Chloride Nanoparticles. Nanomaterials 11(7), 1811, 2021. https://doi.org/10.3390/nano11071811. WOS:000676478400001
3) Marcela-Elisabeta Barbinta-Patrascu; Yulia Gorshkova; Camelia Ungureanu; Nicoleta Badea; Gizo Bokuchava; Andrada Lazea-Stoyanova; Mihaela Bacalum; Alexander Zhigunov; Sanja M. Petrovič, Characterization and Antitumoral Activity of Biohybrids Based on Turmeric and Silver/Silver Chloride Nanoparticles, Materials 14(16), 4726, 2021. WOS:000689574500001; https://doi.org/10.3390/ma14164726
4) Renganathan, S.;Subramaniyan, S.; Karunanithi, N.;Vasanthakumar, P.; Kutzner, A.;Kim, P.-S.; Heese, K. Antibacterial,Antifungal, and Antioxidant Activities of Silver Nanoparticles Biosynthesized from Bauhinia tomentosa Linn. Antioxidants 2021, 10,1959. https://doi.org/10.3390/antiox10121959
The authors must revise this paper and clearly state that it is a review on metal-based nanomaterials.
In many places in the text, they use the names: "NPs", "nanoparticles", "nanomaterials" in general. The authors must clearly specify that this review is focused on metal-based nanomaterials. The authors must revise it in whole text. The title should also be reformulated.
I made many corrections and suggestions in the attached manuscript.
Authors must pay attention to the magenta highlighted words/sentences.
Other suggestions are the following:
1) The Latin words (via, in vivo, in vitro) and Latin names of plant species (Mytilus galloprovincialis) or microorganisms (e.g., E. coli, Staphylococcus aureus, and Pseudomonas aeruginosa) must be written italic.
2) Authors must give a reference for the sentence from Lines 46-48.
3) Lines 67-71: Figure 1: The legend must be reformulated. Replace "Biokinetic" with "biointeraction" or "bioimpact".
4) Line 135: Specify what is Florine-19.
5) Line 137: Replace "NPs 198Au" with "198Au-NPs".
6) Lines 211-212: Reformulate the sentence!
7) Lines 230-231: Reformulate the sentence!
8) Lines 247-249: Revise "mouse brain rose" and reformulate the entire sentence!
9) Line 255" Replace "molecules" with "agents".
10) Figure 2: Replace "Free Radicles" with "Free Radicals" on the image.
11) Figure 3: All the abbreviations must be detailed. For example, COPD! Replace "Acna" with "Acne". Specify the type of nanomaterials!
12) Line 318: "ER" - All the abbreviations must be detailed in text!
13) Lines 354-355: Reformulate!
14) Lines 370-371: Specify the composition of these lipid-nanoparticles! Liposomes are biocompatible! See the recommended bibliographic references mentioned above.
etc.

Author Response
Comment-1: The toxicity of metal-based nanoparticles depends on many factors: preparation method, size and shape of nanoparticles, surface properties, concentration, etc. The authors should make a comparison between the different types of metal-based NPs, from recent studies. In many cases, metal and metal oxide nanoparticles (especially the biosynthesized ones) show antioxidant, antimicrobial, and anticancer activities.
Response-1: We are thankful to the reviewer for supporting us in further improvement of this manuscript. This article focuses on the overall critical aspects of metal-based nanoparticles (mNPs) that impacts the assessment of concerns regarding safety and health. Instead of comparing among the mNPs, we strive to stress their adverse consequences on health. Indeed, we described the information on particle shape, size, and threshold dosages in the article.
Comment-2: The authors stated that: "Furthermore, activation of the complement cascade in response to liposomes and other lipid-based nanoparticles causes hypersensitivity responses and anaphylaxis [125,126]."
But, liposomes are biocompatible and they are frequently used in biomedical fields, in cosmetics, in food industry, etc.!
Response-2: We appreciate the reviewer for supporting us in the further improvement of this manuscript. The purpose of this paper is to discuss about the adverse effects of metal-based nanoparticles on induced cytotoxicity. The focus of this paper will not, however, be liposomes or other lipid-based nanoparticles. Thus, the statement is omitted from the draft.
Comment-3: I recommend updating the bibliography and introducing the following bibliographic references related to metal nanoparticles and liposomes:
1) Alemán, A.; Pérez-García,S.; Fernández de Palencia, P.; Montero, M.P.; Gómez-Guillén,M.d.C. Physicochemical, Antioxidant, and Anti-Inflammatory Properties of Rapeseed Lecithin Liposomes Loading a Chia (Salvia hispanica L.) Seed Extract. Antioxidants 2021, 10, 693. https://doi.org/10.3390/antiox10050693
Not applicable for this article
2) Gorshkova, Y.; Barbinta-Patrascu, M.-E.; Bokuchava, G.; Badea, N.; Ungureanu, C.; Lazea-Stoyanova, A.; Răileanu, M.; Bacalum, M.; Turchenko, V.; Zhigunov, A.; Juszyńska-Gałązka, E. Biological Performances of Plasmonic Biohybrids Based on Phyto-Silver/Silver Chloride Nanoparticles. Nanomaterials 11(7), 1811, 2021.
3) Marcela-Elisabeta Barbinta-Patrascu; Yulia Gorshkova; Camelia Ungureanu; Nicoleta Badea; Gizo Bokuchava; Andrada Lazea-Stoyanova; Mihaela Bacalum; Alexander Zhigunov; Sanja M. Petrovič, Characterization and Antitumoral Activity of Biohybrids Based on Turmeric and Silver/Silver Chloride Nanoparticles, Materials 14(16), 4726, 2021. WOS:000689574500001;
4) Renganathan, S.;Subramaniyan, S.; Karunanithi, N.;Vasanthakumar, P.; Kutzner, A.;Kim, P.-S.; Heese, K. Antibacterial,Antifungal, and Antioxidant Activities of Silver Nanoparticles Biosynthesized from Bauhinia tomentosa Linn. Antioxidants 2021, 10,1959. https://doi.org/10.3390/antiox10121959
Response-3: I'd like to thank you for informing us about the inadvertent error in the manuscript. We have included the suggested reference (no. 2 and 3) in the “Conclusion and further prospective” section “An approach called "green" synthesis of metal-based NPs from plant extracts must be prioritized in hopes of reducing the cytotoxicity [184,185]. Nevertheless, we regret to inform you that we could not uncover any relevant quotations from references 1 and 4.
Comment-4: The authors must revise this paper and clearly state that it is a review on metal-based nanomaterials.
Response-4: We revise and rewrite the sentences throughout the manuscript to improve readability, and comprehension and make it easier to understand. All the points are addressed thoroughly and the necessary changes have been done.
Comment-5: In many places in the text, they use the names: "NPs", "nanoparticles", "nanomaterials" in general. The authors must clearly specify that this review is focused on metal-based nanomaterials. The authors must revise it in whole text. The title should also be reformulated.
Response-5: First and foremost, we express my gratitude for informing us about the inaccuracy in the manuscript's title. We clearly specify the “metal-based nanomaterials” in the text. I'd want to thank you for notifying us of the inadvertent error that has been scribbled in the title of this work. According to the manuscript's prior title, I completely agreed with the reviewer. Indeed, I happily welcome the reviewer's further letter for additional changes to the title.
Comment-6: “Metal-based Nanoparticles and its’ associate influences on Induced Oxidative Stress and Cytotoxicity Cascade as Health Consequences- A Systematic Review”
Response-6: We revise and rewrite the whole manuscript clearly specify the metal-based nanomaterials (mNPs)
Comment-7: I made many corrections and suggestions in the attached manuscript.
Response-7: We carefully considered the comments and tried our best to address every one of them.
Comment-8: Authors must pay attention to the magenta highlighted words/sentences.
Response-8: We addressed the mistake and incorporated it into the revised document.
Other suggestions are the following:
Comment-9: 1) The Latin words (via, in vivo, in vitro) and Latin names of plant species (Mytilus galloprovincialis) or microorganisms (e.g., E. coli, Staphylococcus aureus, and Pseudomonas aeruginosa) must be written italic.
Response-9: We addressed the reviewers' concerns and changes have been done with the Italic font.
Comment-10: 2) Authors must give a reference for the sentence from Lines 46-48.
Response-10: The reference has been included. “The population susceptible to material-based nanoparticles (mNPs) is growing as their application develops. According to the Allied market report (https://www.alliedmarketresearch.com/nanotechnology-market, published in Jul, 2021)”. (Line no. 48-49)
Comment-11: 3) Lines 67-71: Figure 1: The legend must be reformulated. Replace "Biokinetic" with "biointeraction" or "bioimpact".
Response-11: The Figure 1 legend is reformed with “Nanoparticle and its bio interaction”.
Comment-12: 4) Line 135: Specify what is Florine-19.
Response-12: Fluorine-19 is a stable isotope of fluorine and naturally occurring in more than trace quantities, it has nine protons and ten neutrons. This manuscript specifies the metal-based nanomaterials and their consequences, therefor we remove the Florine-19 statement from the article.
Comment-13: 5) Line 137: Replace "NPs 198Au" with "198Au-NPs".
Response-13: We replaced "NPs 198Au" with "198Au-NPs".
Comment-14: 6) Lines 211-212: Reformulate the sentence!
Response-14: We modified the sentence (Line no. 341-342). “The dose-dependent Ag NPs-induced toxicity experiment was carried out in zebrafish embryos [111]”.
Comment-15: 7) Lines 230-231: Reformulate the sentence!
Response-15: We reformed the sentence (Line no. 375-377) as “It has been demonstrated that these NPs interact directly with the liver and, as they migrate through the organ systems, might cause synaptic damage and neurotransmitter dysfunction called neurotoxicity [2,128]”.
Comment-16: 8) Lines 247-249: Revise "mouse brain rose" and reformulate the entire sentence!
Response-16: The whole sentence has been modified as “After being exposed to TiO2 NPs, the concentration of Ti in the mouse brain elevated, causing an increase in ROS generation, inhibition of antioxidant activities in hippocampal regions, and an increase in the proportion of apoptotic cells [138]”. (Line no. 392-394)
Comment-17: 9) Line 255" Replace "molecules" with "agents".
Response-17: The "molecules" were replaced with "agents". (Line no. 400)
Comment-18: 10) Figure 2: Replace "Free Radicles" with "Free Radicals" on the image.
Response-18: Figure number 2 has been changed to Figure 3, the spelling mistake "Free Radicles" replaced with "Free Radicals" (Line no. 421)
Comment-19: 11) Figure 3: All the abbreviations must be detailed. For example, COPD! Replace "Acna" with "Acne". Specify the type of nanomaterials!
Response-19: The figure number is modified with 2. We have specified all the abbreviations in the text.
Moreover, we have discussed the effects of various types of the metals based nanoparticles on the organs (Section 3.1. Line no. 182-301)
Comment-20: 12) Line 318: "ER" - All the abbreviations must be detailed in text!
Response-20: We have specified the abbreviations in the text endoplasmic reticulum stress. (Line no. 458)
Comment-21: 13) Lines 354-355: Reformulate!
Response-21: The sentence has been modified. Excessive ROS production and oxidative stress can activate nuclear respiratory factor (NRF) 1 and 2 via AKT (Protein Kinase B), estrogen receptor (ESR) 1, and alter proteins function implicated in the stress response pathway such as SODs, CAT, GR, GPXs, PDI, and PRDXs. (Line no. 501-504)
Comment-22: 14) Lines 370-371: Specify the composition of these lipid-nanoparticles! Liposomes are biocompatible! See the recommended bibliographic references mentioned above.
etc.
Response-22: I'd like to thank you for informing us about the inadvertent error in the manuscript. We realized that the sentence is irrelevant to the section, so we have deleted it and rewrote the section.
Reviewer 2 Report
The balance between the advantages and disadvantages of the abundant use of nanomaterials and nanoparticles has been discussed for many years. The idea, the content of the article and the topics covered are well chosen and discussed in the presented manuscript. However, there are still some concerns that should be taken into consideration.
Major remarks:
1/ nomenclature used by the authors is confusing and must be unified - in the title as well as in different parts of article, the Authors mention "nanomaterials", however the body of the manuscript concerns rather metal nanoparticles. It should be precisely indicated what kind of nanostructures are mentioned in different parts.
2/ chapter 3. "Nanomaterials induced physio-pathological effects on health" is very short, however the Authors continue writing about in vivo effects in the 4th chapter (entitled "Cytotoxicity and cellular damage"), and present some outcomes in Figure 3 at the end of the manuscript. It should be literally indicated which effects of NPs influence are on the whole body and which are on the cellular level.
3/ English revision is required
Minor remarks:
1/ the abbreviations must be explained when used for the first time in the text. e.g. line 65 (NPs)
2/ correct the use of capital letters in some words, e.g. line 46 "Nanotechnology market" or line 72 "Nanotoxicology"
3/ correct the subscript in chemical formulas, e.g. lines 102, 104, 123, 245, 247
Author Response
Major remarks:
Comment-1: 1/ nomenclature used by the authors is confusing and must be unified - in the title as well as in different parts of article, the Authors mention "nanomaterials", however the body of the manuscript concerns rather metal nanoparticles. It should be precisely indicated what kind of nanostructures are mentioned in different parts.
Response-1: We express my gratitude for informing us about the inaccuracy in the manuscript. This article focuses on the overall critical aspects of metal-based nanoparticles (mNPs) on the health. We addressed the reviewers' concerns. Hence, we reformulated the title and clearly specify the same in the manuscript.
Comment-2: 2/ chapter 3. "Nanomaterials induced physio-pathological effects on health" is very short, however the Authors continue writing about in vivo effects in the 4th chapter (entitled "Cytotoxicity and cellular damage"), and present some outcomes in Figure 3 at the end of the manuscript. It should be literally indicated which effects of NPs influence are on the whole body and which are on the cellular level.
Response-2: We are thankful to the reviewer for supporting us in further improvement of this manuscript. More specifically, we addressed Section 3 and added it to the paper. The amended manuscript's figure number 3 and replaced with figure number 2. Moreover, we have discussed the effects of various types of metals-based nanoparticles on the organs (Section 3.1. Line no. 182-301).
Comment-3 3/ English revision is required
Response-3: We rectified the mistake and integrated it into the document. To improve readability and comprehension, we revise and rewrite the entire manuscript.
Minor remarks:
Comment-4: 1/ the abbreviations must be explained when used for the first time in the text. e.g. line 65 (NPs)
Response-4: The corrections have been included.
Comment-5: 2/ correct the use of capital letters in some words, e.g. line 46 "Nanotechnology market" or line 72 "Nanotoxicology"
Response-5: The correction has been included. (Line no 50)
Comment-6: 3/ correct the subscript in chemical formulas, e.g. lines 102, 104, 123, 245, 247
Response-6: All chemical formulas have been reviewed and updated accordingly.
Reviewer 3 Report
The manuscript entitled “Nanomaterials and associated influence on Induced Cytotoxicity and Oxidative Stress Cascade as Health Consequences” aimed to investigate the mechanisms and cellular signalling cascades of metal nanoparticle-induced oxidative stress and their relevant health consequences.
Page 2, line 51 – please rephrase “…followed by the re-distributed to the major organs, potentially causing tissue damage.”
I would suggest to provide section with the Search strategy. How did authors select papers to be included in this review? Which data bases were searched and with what keywords. What were including/excluding criteria/parameters? This is very important for a good literature overview. Besides, what is the type of this review paper… narrative, integrative, systematic, scoping etc.?
Minor remarks:
Abstract – please put full word for NP’s in the second sentence
Abstract – remove one dot at the end of the last sentence
Page 2, line 51 - please put full word for NP’s on the first mention
Page 3, line 76 – use small “r” in Reactive oxygen species
In vitro and in vivo should be in Italic
Page 3, line 123 – use subscript in Fe2O3
Page 4, line 157 – ROS as an abbreviation was already introduced; page 5 and 6… the same
Page 5, line 185 – change In-vivo to In vivo
Author Response
The manuscript entitled “Nanomaterials and associated influence on Induced Cytotoxicity and Oxidative Stress Cascade as Health Consequences” aimed to investigate the mechanisms and cellular signalling cascades of metal nanoparticle-induced oxidative stress and their relevant health consequences.
Comment-1: Page 2, line 51 – please rephrase “…followed by the re-distributed to the major organs, potentially causing tissue damage.”
Response-1: We have rewritten the sentence. “The mNPs are frequently consumed through food, drink, air, cosmetics, or pharmaceuticals and absorbed by the body before being re-distributed to the major organs, potentially causing tissue damage”. (Line no. 53-56)
Comment-2: I would suggest to provide section with the Search strategy. How did authors select papers to be included in this review? Which data bases were searched and with what keywords. What were including/excluding criteria/parameters? This is very important for a good literature overview. Besides, what is the type of this review paper… narrative, integrative, systematic, scoping etc.?
Response-2: I'd like to thank you for informing us about the inadvertent error in the manuscript. We rewrite sentences throughout the manuscript to improve readability, comprehension, and understanding. We also added more explanations to rephrase the objectives and conclusion of the study. The title has been modified as “Metal-based Nanoparticles and its’ associate influences on Induced Oxidative Stress and Cytotoxicity Cascade as Health Consequences- A Systematic Review”
Minor remarks:
Comment-3: Abstract – please put full word for NP’s in the second sentence
Response-3: The full word is included in the manuscript. The modified sentence is “The lack of information on bioimpacts, as well as the absence of consistent standards, are the limitations of uses the material-based nanoparticles (mNPs) for existing applications”. (Line no. 24-26).
Comment-4: Abstract – remove one dot at the end of the last sentence
Response-4: The dot is removed from the last sentence in the Abstract.
Comment-5: Page 2, line 51 - please put full word for NP’s on the first mention
Response-5: The whole sentence has been modified.
Comment-6: Page 3, line 76 – use small “r” in Reactive oxygen species
Response-6: The correction has been done in the entire manuscript.
Comment-7: In vitro and in vivo should be in Italic
Response-7: We addressed the reviewers' concerns and changes have done to Italic font.
Comment-8: Page 3, line 123 – use subscript in Fe2O3
Response-8: All chemical formulas have been reviewed and updated accordingly.
Comment-9: Page 4, line 157 – ROS as an abbreviation was already introduced; page 5 and 6… the same
Response-9: We have corrected all the reparation of the “ROS” abbreviation in the manuscript
Comment-10: Page 5, line 185 – change In-vivo to In vivo
Response-10: It has been corrected from In vitro to in vivo (Line no. 314)
Round 2
Reviewer 1 Report
The manuscript antioxidants-2241392 has been improved and can be published after a minor revision. Please find attached the manuscript with new corrections/suggestions. Authors must pay attention to the magenta highlighted words/sentences. Briefly, I suggested the following:
1) Figures must be renumbered.
2) Lines 259-264: This sentence must be reformulated.
3) Lines 345-356: This paragraph should be deleted because it is not the subject of this review.
4) Line 368: Reformulate: "cell death elevation of inflammation-related genes".
5) Line 430: Replace "The function of mNPs" with "The mNPs could function".

Author Response
Reviewer 1
Comment: The manuscript antioxidants-2241392 has been improved and can be published after a minor revision. Please find attached the manuscript with new corrections/suggestions. Authors must pay attention to the magenta-highlighted words/sentences. Briefly, I suggested the following:
Response: I'd like to thank you for again informing us about the inadvertent error in the manuscript. We have included the suggestions in the revised manuscript.
Commnet-1: 1) Figures must be renumbered.
Response-1: The figure numbers are modified and arranged accordingly.
Commnet-2: 2) Lines 259-264: This sentence must be reformulated.
Response-2: We addressed the reviewer’s note and remove the sentence from the revised document.
Commnet-3: 3) Lines 345-356: This paragraph should be deleted because it is not the subject of this review.
Response-3: We remove the paragraph from the manuscript.
Commnet-4: 4) Line 368: Reformulate: "cell death elevation of inflammation-related genes".
Response-4: We revise and rewrite the sentence in line number 358-359.
Commnet-5: 5) Line 430: Replace "The function of mNPs" with "The mNPs could function".
Response-5: The change has been included in line no 421.
Reviewer 2 Report
The Authors have responded my questions and improved their manuscript significantly. The main drawback in the corrected manuscript is the title - too long, and quite awkward. I leave the final decision concerning the manuscript title to the Editors.
Minor corrections:
p. 1, line 40 - the abbreviation NPs is used for the first time so should be explained
p.2, line 47 - the abbreviation mNPs is used for the first time so should be explained
p.5 - figure 2A should be numbered as 2, and figure 2B as 3
p.10, 11 and further - please number the subchapters consequently 4.1, 4.2 and so on
Author Response
Reviewer 2
Comment-1: The Authors have responded my questions and improved their manuscript significantly. The main drawback in the corrected manuscript is the title - too long, and quite awkward. I leave the final decision concerning the manuscript title to the Editors.
Response-1: We appreciate the reviewer for supporting us in the further improvement of this manuscript. We revise and rewrite the title as “Metal-based Nanoparticles and Its Relevant Consequences on Cytotoxicity Cascade and Induced Oxidative Stress - A Systematic Review”.
Indeed, I happily welcome the reviewer's further letter for additional changes to the title.
Minor corrections:
Comment-2: p. 1, line 40 - the abbreviation NPs is used for the first time so should be explained
Response-2: We addressed the reviewers' concerns and changes have been made in line no. 40.
Comment-3: p.2, line 47 - the abbreviation mNPs is used for the first time so should be explained
Response-3: We clearly specify the abbreviation of metal-based nanomaterials (mNPs) line no. 47.
Comment-4: p.5 - figure 2A should be numbered as 2, and figure 2B as 3
Response-4: The figure numbers are modified and arranged accordingly.
Comment-5: p.10, 11 and further - please number the subchapters consequently 4.1, 4.2 and so on
Response-5: The sequential numbering of the subchapter has been rearranged.